# PHYSICS INFORMED NEURALLY CONSTRUCTED ODE NETWORKS (PINECONES)

## ABSTRACT

Recently, there has been a growing interest in using neural networks to approximate the solutions of partial differential equations (PDEs). Physics-informed neural networks (PINNs) have emerged as a promising framework for parameterizing PDE solutions using deep neural networks. However, PINNs often rely on memory-intensive optimizers to attain reasonable accuracy and can encounter training difficulties due to issues such as stiffness in the gradient flow of the loss. To address these challenges, we propose a novel network architecture that combines neural ordinary differential equations (ODEs) with physics-informed constraints in the loss function. In this approach, the dynamics within a neural ODE are expanded to include a system of ODEs whose solution provides the partial derivatives governing our PDE system. We call this architecture PINECONEs: physics-informed neurally constructed ODE networks. We evaluate the approach using simple but canonical PDEs from the literature to illustrate its potential. Our results show that training requires fewer iterations than previous approaches to achieve higher accuracy when using first-order optimization methods.

## 1 INTRODUCTION

Partial differential equations (PDEs) play a vital role in mathematical modelling and simulation, but most PDEs lack analytical solutions and must be numerically approximated. Although many tools exist for numerically solving a variety of PDEs, many problems of scientific interest remain computationally intractable or require significant simplification.

The fields of machine learning and scientific computing have recently undergone a synergistic interplay. This newly emerging area has been coined "scientific machine learning." Neural networks, while expensive and time-consuming to train, are incredibly fast to evaluate, making them ideal for applications where simulations are infeasible or computationally costly. Hybrid modelling frameworks that incorporate neural networks into scientific modelling problems have yielded impressive results Karniadakis et al. (2021b).

One successful approach is Physics Informed Neural Networks (PINNs) (Raissi et al., 2019a), which have been used to model multi-scale and multi-physics phenomena, solve high-dimensional systems of PDEs, and combine incomplete mechanistic understanding with data (Karniadakis et al., 2021a). However, certain types of problems remain open challenges for PINNs. For instance, PINNs may struggle to train for multi-scale problems or problems that exhibit multiple frequencies in their solutions. To achieve reasonable accuracy, PINNs are often trained using the LBFGS optimizer (Zhu et al., 1997), a quasi-Newton method which is more computationally costly and memory intensive than first-order methods.

In this work, we adapt neural ODEs so that they may be used in a PINN. We demonstrate that, on some simple PDE benchmark problems, our method is able to attain lower error with less training than the original PINN approach when using a first-order optimizer.

## 1.1 Background

Consider a system of partial differential equations (PDEs),

$$\mathcal{F}u = f \ \text{ in } \ \Omega \tag{1}$$
$$\mathcal{B}u = g \ \text{ in } \ \partial\Omega$$

where $\mathcal{F}(\cdot)$ is a differential operator, $\mathcal{B}(\cdot)$ is a boundary value operator, $\Omega \subset \mathbb{R}^n$, $\partial\Omega$ is an appropriately defined boundary, and $f$ and $g$ are known functions in suitable function spaces; $u$ is the unknown solution we aim to approximate. Note that $g \in \partial\Omega$ will include initial conditions for time-dependent systems. Finding $u$ is equivalent to finding the root of the residual operator

$$\mathcal{R}(u) = \|\mathcal{F}u - f\|^p + \|\mathcal{B}u - g\|^p \tag{2}$$

for suitable norms with $p > 0$.

One approach for approximating solutions to Eq. (1) is to use a collocation method. In a collocation method, a candidate solution is sought among a class of functions in some finite dimensional space to best satisfy the differential equation at a set of chosen points referred to as the collocation points. For example, in pseudo-spectral methods (Shen et al., 2011), the family of candidate solutions is trigonometric polynomials of some fixed order, parameterized by their coefficients. The goal is to find the coefficients that best satisfy the differential equation on some collocation points. Another space of functions that can be used to generate a family of candidate solutions to Eq. (1) is the set of neural networks with a given architecture. For readability, the class of neural networks under consideration in this paper is briefly described below.

**Feedforward Neural Networks** A feedforward fully connected neural network is a mapping that takes input $x$ through a composition of affine maps with nonlinear functions applied componentwise, referred to as activation functions. Such a mapping can be written as

$$F_\theta(x) = C_L \circ \sigma_L \circ C_{L-1} \circ \cdots \circ \sigma_1 \circ C_1(x) \quad \text{where} \quad C_k(z) = W_k z + b_k, \tag{3}$$
$$W_k \text{ is a } n_k \times n_{k-1} \text{ matrix, } b_k \in \mathbb{R}^{n_k}, \text{ and } \theta = \{W_k, b_k : 1 \le k \le L\}.$$

At each $k$, $\sigma_k(C_k(\cdot))$ is referred to as a layer, and the number of layers $L$ is called the neural network's depth. The size of layer $n_k$ is referred to as the width of the network at layer $k$, with $n_0$ the dimension of the input $x$. We will stick to the class of neural networks where the scalar nonlinear activation functions are smooth and the same for each layer, $\sigma_k = \sigma \ \forall k$.

## 1.2 Neural Ordinary Differential Equations

In Neural Ordinary Differential Equations (Neural ODEs), a neural network parameterizes the vector field of an initial value problem (IVP) for a system of ordinary differential equations (Chen et al., 2018). Concretely, consider the IVP:

**Neural Ordinary Differential Equation**

$$\begin{cases} \frac{d\mathbf{z}(\tau,\mathbf{z_0})}{d\tau}\Big|_{(\tau,\mathbf{z_0})} = \mathbf{F}_\theta(\mathbf{z}(\tau,\mathbf{z}_0),\tau) \\ \mathbf{z}(0,\mathbf{z}_0) = \mathbf{z}_0 \end{cases} \tag{4}$$

| | |
|---|---|
| $\mathbf{z}_0$ | $\mathbb{R}^m$ |
| $\mathbf{z}(\tau,\mathbf{z}_0)$ | $\mathbb{R}\times\mathbb{R}^m \to \mathbb{R}^m$ |
| $\mathbf{F}_\theta(\mathbf{z}(\tau,\mathbf{z}_0),\tau)$ | $\mathbb{R}^m\times\mathbb{R}\to\mathbb{R}^m$ |

where the vector field $\mathbf{F}_\theta$ is described by a neural network Eq. (3)). The aim is to optimize parameters $\theta$ so that the solution to Eq. (4) best describes the mapping $\mathbf{x} \mapsto \mathbf{u}$ where $\mathbf{z}_0 = \mathbf{x}$ and $\mathbf{z}(1,\mathbf{z}_0) \approx \mathbf{u}$. Given any scalar-valued loss function $\mathcal{L}$ (e.g. mean squared error), this is achieved by minimization of the loss

$$\mathcal{L}\big(\mathbf{z}(1,\mathbf{z}_0),\mathbf{u}\big) = \mathcal{L}\left(\left[\mathbf{z}(0,\mathbf{z}_0) + \int_0^1 \mathbf{F}_\theta\left(\mathbf{z}(s,\mathbf{z}_0),s\right)ds\right],\mathbf{u}\right)$$
$$= \mathcal{L}\big(\text{ODESolve}(\mathbf{z}(0,\mathbf{z}_0),\mathbf{F}_\theta,0,1,\theta),\mathbf{u}\big).$$

**Augmented Neural ODEs**   Dupont et al. (2019) showed that because the flows of Neural ODEs are homeomorphisms, there are functions that they cannot represent. To address this, they proposed Augmented Neural ODEs (ANODEs). In an ANODE, the dimension of the Neural ODE is augmented by appending zeros to the input space. For example, if the original ODE in a Neural ODE system has dimension $d_p$ with input data $(\mathbf{x}_1, \ldots, \mathbf{x}_{d_p})$, simply augment with zeros $(\mathbf{x}_1, \ldots, \mathbf{x}_{d_p}, \mathbf{0}, \ldots, \mathbf{0}_{d_p+d_a})$, to get a $d_a$-augmented ANODE. Augmenting the dimension allows the Neural ODE to represent a larger class of functions and often improves computational efficiency by reducing the number of function evaluations used by the numerical ODE solver during training. In this work, we elect to use ANODEs for all the Neural ODE implementations due to their increased representational range.

## 1.3 Physics Informed Neural Networks

In a PINN, a fully connected feedforward neural network approximates $u \approx F_\theta(x)$ from Eq. (3) (Raissi et al., 2019a). This network, which we denote as $u_\theta = F_\theta$, is parameterized by $\theta \in \mathbb{R}^m$. Substituting $u$ with $u_\theta$ in Eq. (2), the objective is to find the optimal $\theta$ values that minimize $\mathcal{R}(u_\theta)$ on a set of collocation points sampled via a Monte Carlo approach or chosen with other criteria. Derivatives of $u_\theta$ can be readily found via automatic differentiation. In practice, the norms in (Eq. (2)) are replaced by Euclidean norms or an appropriately defined quadrature approximation of the integral.

## 1.4 Main Contribution and Broader Impact

We extend the class of neural networks that can be used with a physics-informed loss to include Neural ODEs. We call our approach Physics Informed Neurally Constructed ODE Networks (PINECONEs). Neural ODEs have properties that are useful for physics-informed applications, such as the ability to handle irregularly sampled time-series data. Furthermore, the diverse adaptations of the Neural ODE framework, such as Hamiltonian neural networks, stochastic neural ODEs, and others, open up new avenues for physics-informed machine learning. Conversely, our approach provides a method for computing the sensitivity of a Neural ODE to input data in a memory-efficient manner, which may be of broader interest for a range of tasks involving Neural ODEs.

Our results show that training requires fewer iterations than ordinary PINNs to achieve higher accuracy when using first-order optimization methods. This is particularly important when incorporating real-world data into the PINN framework, where optimizers like LBFGS may not scale well with larger data sets. Similarly, first-order optimizers are more suitable for problems involving high-dimensional PDEs.

## 1.5 Related Work

- Many works address the accuracy and training difficulties in PINNs; see (Müller & Zeinhofer, 2023) for an overview. Wu et al. (2023) study strategies for sampling the collocation points used in training to improve training. van der Meer et al. (2022) focus on adaptively weighing different components of the PINN loss. Wang et al. (2022a) forces a PINN to respect causality by modifying the PINN loss to enforce temporal order in time-dependent problems. Wang et al. (2021) suggests a specialized architecture to improve training. All of these approaches are independent of the work in this proposal. Indeed, a PINECONE may be combined with any of the strategies listed above.

- Various optimization strategies have been suggested for PINN training (Zeng et al., 2022; Davi & Braga-Neto, 2022; Müller & Zeinhofer, 2023). However, these can also be combined with PINECONEs by using any of them for training. Moreover, in this work, rather than replacing first-order optimization methods, we embrace them due to their scalability.

- In both Lee & Parish and Rackauckas et al. (2021), PDEs are transformed into an ODE system and solved with Neural ODEs. In Rackauckas et al. (2021), a PDE is analytically transformed into a system of ODEs and then solved with a Neural ODE, while in Lee & Parish, the PDE is discretized in space and a left continuous in time, then a Neural ODE is used on the resulting ODE system. These approaches most closely resemble PINECONEs. However, PINECONEs are unique in that they provide a continuous solution and use an ODE to approximate a PDE rather than reducing a PDE problem to an ODE problem. Another novel feature of PINECONEs is the use of forward sensitivity equations to compute derivatives.

## 2 PINECONEs

In a Neural ODE, the neural network must satisfy a dynamical system by construction. This observation motivates using Neural ODEs as a family of candidate functions for the numerical solution of Eq. (1) via a Physics informed collocation approach. A Neural ODE $\mathbf{z}(1, \mathbf{z}_0)$ is mapped to the correct dimension for the PDE problem via a linear transformation $\mathbf{A}$ and substituted into Eq. (2). A suitable discretization of $\mathcal{R}(\mathbf{A}\mathbf{z})$ is minimized at the set of collocation points $\mathbf{z}_0 = \mathbf{x}$. See Section 2.3 for details on $\mathbf{A}$.

The difficulty in using Neural ODEs to approximate $u$ lies in how to best go about taking derivatives of $\mathbf{z}(1, \mathbf{z}_0) = u_\theta \approx u$ with respect to $\mathbf{z}_0$. In a Neural ODE, the training data is a set of initial conditions. For a Physics informed training task performed with a Neural ODE, these initial conditions represent the set of spatial/temporal coordinates of the PDE. Naively using auto-differentiation on $\mathbf{z}$ requires differentiating through all the operations of the ODE solver employed in the forward pass. For example, consider backpropagation using the simple first-order one-dimensional differential operator $\mathcal{F} = \frac{\partial}{\partial x} + \frac{\partial}{\partial t}$. Derivatives of the loss are

$$
\begin{aligned}
\frac{\partial \mathcal{L}}{\partial \theta} &= \frac{\partial}{\partial \theta}\left(\mathcal{L}\left(\mathcal{F}\left(\mathbf{A}\left[\mathbf{z}(0, \mathbf{z}_0) + \int_0^1 \mathbf{F}_\theta\left(\mathbf{z}(s, \mathbf{z}_0), s\right)\, ds\right]\right)\right)\right) \\
&= \frac{\partial}{\partial \theta}\left(\mathcal{L}\left(\frac{\partial}{\partial x}\left(\mathbf{A}\left[\text{ODESolve}\left(\mathbf{z}(0, \mathbf{z}_0), \mathbf{F}_\theta, 0, 1, \theta\right)\right]\right)\right.\right. \\
&\qquad\qquad \left.\left. + \frac{\partial}{\partial t}\left(\mathbf{A}\left[\text{ODESolve}\left(\mathbf{z}(0, \mathbf{z}_0), \mathbf{F}_\theta, 0, 1, \theta\right)\right]\right)\right)\right).
\end{aligned}
$$

For a large set of training data, or for an ODE solve that requires small time steps, this operation becomes memory intensive and costly. To avoid this issue, it is possible to solve an extended system of ODEs that returns not just $\mathbf{z}(1, \mathbf{z}_0)$, but also its partial derivatives with respect to the input data $\mathbf{z}_0$.

### 2.1 FIRST-ORDER PINECONE

Introducing $S_1(\tau, \mathbf{z}_0) := \nabla\big|_{(\tau, \mathbf{z}_0)} \mathbf{z}(\tau, \mathbf{z}_0) \in \mathbb{R}^m \times \mathbb{R}^m$ we can extend the original formulation of a Neural ODE. We call the extended coupled system a first-order PINECONE.

**1st order PINECONE**

$$
\begin{cases}
\frac{d\mathbf{z}}{d\tau}\big|_{(\tau, \mathbf{z}_0)} = \mathbf{F}_\theta(\mathbf{z}, \tau) \\
\frac{dS_1}{d\tau}\big|_{(\tau, \mathbf{z}_0)} = \nabla\big|_{(\tau, \mathbf{z}_0)} \mathbf{F}_\theta(\mathbf{z}, \tau) S_1 \\
\mathbf{z}(0, \mathbf{z}_0) = \mathbf{z}_0 \\
S_1(0, \mathbf{z}_0) = \mathbf{I}
\end{cases}
\tag{5}
$$

| | |
|---:|:---:|
| $\mathbf{z}_0$ | $\mathbb{R}^m$ |
| $\mathbf{I}$ | $\mathbb{R}^m \times \mathbb{R}^m$ |
| $\mathbf{z}(\tau, \mathbf{z}_0)$ | $\mathbb{R} \times \mathbb{R}^m \to \mathbb{R}^m$ |
| $S_1(\tau, \mathbf{z}_0)$ | $\mathbb{R}^m \times \mathbb{R}^m$ |
| $\mathbf{F}_\theta(\mathbf{z}(\tau, \mathbf{z}_0), \tau)$ | $\mathbb{R}^m \times \mathbb{R} \to \mathbb{R}^m$ |
| $\nabla\big|_{(\tau, \mathbf{z}_0)} \mathbf{F}_\theta(\mathbf{z}(\tau, \mathbf{z}_0), \tau)$ | $\mathbb{R}^m \times \mathbb{R}^m$ |

Solutions for a first-order PINECONE system are the original Neural ODE solution $\mathbf{z}$ together with the Jacobian of $\mathbf{z}$ with respect to the Neural ODE input data $\mathbf{z}_0$. In other words, in solving this new Neural ODE system, we can access the Neural ODE approximation to the solution and any first-order partial derivative of the Neural ODE approximation.

### 2.2 SECOND-ORDER PINECONE

Setting $S_2(\tau, \mathbf{z}_0) := \nabla\big|_{(\tau, \mathbf{z}_0)} S_1 \in \mathbb{R}^m \times \mathbb{R}^m \times \mathbb{R}^m$ gives a second order PINECONE:

**2nd order PINECONE**

$$
\begin{cases}
\frac{d\mathbf{z}}{d\tau}\big|_{(\tau,\mathbf{z}_0)} = \mathbf{F}_\theta(\mathbf{z},\tau) \\
\frac{dS_1}{d\tau}\big|_{(\tau,\mathbf{z}_0)} = \nabla\big|_{(\tau,\mathbf{z}_0)}\mathbf{F}_\theta(\mathbf{z},\tau)S_1 \\
\frac{dS_2}{d\tau}\big|_{(\tau,\mathbf{z}_0)} = \nabla\big|_{(\tau,\mathbf{z}_0)}\Big(\nabla\big|_{(\tau,\mathbf{z}_0)}\mathbf{F}_\theta(\mathbf{z},\tau)\Big)\otimes S_1 \\
\qquad\qquad\quad + \nabla\big|_{(\tau,\mathbf{z}_0)}\mathbf{F}_\theta(\mathbf{z},\tau)\otimes S_2 \\
\mathbf{z}(0,\mathbf{z}_0) = \mathbf{z}_0 \\
S_1(0,\mathbf{z}_0) = \mathbf{I} \\
S_2(0,\mathbf{z}_0) = \mathbf{0}
\end{cases}
\tag{6}
$$

| | |
|---|---|
| $\mathbf{z}_0$ | $\mathbb{R}^m$ |
| $\mathbf{I}$ | $\mathbb{R}^m \times \mathbb{R}^m$ |
| $\mathbf{0}$ | $\mathbb{R}^m \times \mathbb{R}^m \times \mathbb{R}^m$ |
| $\mathbf{z}(\tau,\mathbf{z}_0)$ | $\mathbb{R}\times\mathbb{R}^m\to\mathbb{R}^m$ |
| $S_1(\tau,\mathbf{z}_0)$ | $\mathbb{R}^m\times\mathbb{R}^m$ |
| $S_2(\tau,\mathbf{z}_0)$ | $\mathbb{R}^m\times\mathbb{R}^m\times\mathbb{R}^m$ |
| $\mathbf{F}_\theta(\mathbf{z}(\tau,\mathbf{z}_0),\tau)$ | $\mathbb{R}^m\times\mathbb{R}\to\mathbb{R}^m$ |
| $\nabla\big|_{(\tau,\mathbf{z}_0)}\mathbf{F}_\theta(\mathbf{z}(\tau,\mathbf{z}_0),\tau)$ | $\mathbb{R}^m\times\mathbb{R}^m$ |
| $\nabla\big|_{(\tau,\mathbf{z}_0)}\Big(\nabla\big|_{(\tau,\mathbf{z}_0)}\mathbf{F}_\theta(\mathbf{z}(\tau,\mathbf{z}_0),\tau)\Big)$ | $\mathbb{R}^m\times\mathbb{R}^m\times\mathbb{R}^m$ |

whose solutions are $\mathbf{z}$ and all first and second-order derivatives of $\mathbf{z}$ with respect to $\mathbf{z}_0$. The ODE system may be extended as many times as desired by repeating the process outlined above. Details of the derivations of first and second-order PINECONEs can be found in Appendix B.

**A Note on the tensor product:** The $ijk$-th entry of the product is given by

$$
\left[\frac{dS_2}{d\tau}\big|_{(\tau,\mathbf{z}_0)}\right]_{ijk} = \left(\Big[\nabla\big|_{(\tau,\mathbf{z}_0)}\Big(\nabla\big|_{(\tau,\mathbf{z}_0)}\mathbf{F}_\theta(\mathbf{z},\tau)\Big)\Big]_{[i,:,:]}\cdot[S_1]_{[:,k]}\right)\cdot[S_1]_{[:,j]}
$$
$$
+ [S_2]_{[:,j,k]}\Big[\nabla\big|_{(\tau,\mathbf{z}_0)}\mathbf{F}_\theta(\mathbf{z},\tau)\Big]_{[i,:]}.
$$

This can be verified by computing the vector-valued Hessian of $\mathbf{z}$ component-wise.

## 2.3 Dimension reduction:

Because a Neural ODE is an ODE system, the input dimensions must match the output dimensions. Using a simple affine transformation of $\mathbf{z}$, $w(\mathbf{z}) := \mathbf{Az}(\tau,\mathbf{z}_0) + \mathbf{b}$, we can use a Neural ODE to approximate any mapping from $\mathbb{R}^m$ to $\mathbb{R}^n$ with $m \neq n$. In all the experiments in this paper, the output layer of the PINECONE is mapped to the appropriate size using the linear transformation $\mathbf{A} = \frac{1}{d_p+d_a}\mathbf{1} \in \mathbb{R}\times\mathbb{R}^{d_p+d_a}$, (with $\mathbf{b} = 0$). This transformation averages the outputs of the neural ODE to map them to the appropriate space. To illustrate, given a one-dimensional PDE on a space-time domain and any fixed $(x,t)$, a two-augmented PINECONE approximates $u$ via the relationship

$$
u(x,t) \approx \left(\mathbf{A}\left[\mathbf{z}(0,\mathbf{z}_0) + \int_0^1 \mathbf{F}_\theta\left(\mathbf{z}(s,\mathbf{z}_0),s\right)\,ds\right]\right)
$$
$$
= \frac{1}{4}[1\,1\,1\,1]\left[[x\,t\,0\,0]^T + \int_0^1 \mathbf{F}_\theta\left(\mathbf{z}\left(s,[x\,t\,0\,0]^T\right),s\right)ds\right].
$$

## 3 Experimental Results

The code for all experiments will be made available at the time of publication. The performance of PINECONEs is evaluated on two canonical one-dimensional test problems: the transport equation and Burger's equation. These are benchmarks that have been studied in previous PINN literature (Raissi et al., 2019b; Krishnapriyan et al., 2021; Lee & Parish). The transport equation models transport phenomena and is foundational for a variety of applications, while Burger's equation is the canonical prototype for shock-forming conservation laws, which typical numerical methods struggle with. PINECONEs are compared against a standard PINN approximation on these two test problems.

To ensure a fair comparison, the neural networks used in the PINN and PINECONE approximations have been made as similar as possible. In all experiments, both networks have the same number of layers and identical widths, use the same activation function and are initialized in the same way. We use Swish activations and either He or Orthogonal initialization (He et al., 2015; Saxe et al., 2014).Additionally, the same optimization scheme with the same learning rate is used in training.

The only difference between them is the size of the input data and that of the output layer. In a PINN, the size of the input data corresponds to the dimension of the domain of the PDE, which we denote $d_p$. The output layer is the size of the PDE dimension; in our examples, this equals one. In an ANODE, the input dimension is of size $d_p + d_a$ and must match the size of the output layer. A linear transformation averages the contributions of the output layer (see section Section 2.3), taking them to the PDE's dimension.

**Implementation Details** All of the code is implemented in Python using the Jax library (Bradbury et al., 2018). The neural networks are built using the Equinox library (Kidger & Garcia, 2021). The PINECONEs use the Diffrax library in both the ODE solve for the forward pass and the ODE solve of the adjoint system used in backpropagation (Kidger, 2021). The Optax library is used for the optimization (Babuschkin et al., 2020). All experiments were run with Google Colaboratory using an NVIDIA T4 GPU. The Jacobian-matrix product in Eq. (5) and the tensor product in Eq. (6) are computed as vectorized Jacobian-vector and Hessian-vector products, respectively, for computational efficiency. The code was verified on an artificial problem to ensure the accuracy of the implementation; details of this process are reported in Appendix A.

## 3.1 THE TRANSPORT EQUATION

As a first test problem, we consider the one-dimensional transport equation with periodic boundary conditions and a sine initial condition,

$$\frac{\partial u}{\partial t} + c\frac{\partial u}{\partial x} = 0, \quad \text{for } (x,t) \in \Omega = [0, 2\pi] \times [0, 1], \tag{7}$$
$$u(0,t) = u(2\pi, t), \quad \text{for } x \in \partial\Omega = \{0, 2\pi\},$$
$$u(x, 0) = \sin(x).$$

The constant $c$ represents the speed at which the initial condition is transported. The analytical solution to this problem is $u(x,t) = \sin(x - ct)$. Krishnapriyan et al. (2021) varied the parameter $c$ on a fixed time window and showed that increasing $c$ leads to increased training difficulty for a PINN. However, because increasing $c$ corresponds to a change in the time scale, these training difficulties should disappear under an appropriate rescaling; thus, we take $c = 1$ and compare the performance of a PINN to that of a PINECONE.

For each method, the neural network has eight hidden layers, all with width twenty. The PINECONE's training data is augmented with three zeros, as described in Section 1.2. The networks are trained for 60,000 iterations utilizing the ADAM optimizer (Kingma & Ba, 2017). The learning rate begins at $1e^{-4}$ and is reduced to $1e^{-5}$ after 7,000 iterations. We stick to the default hyperparameters for ADAM provided by Optax. For clarity, we briefly describe the physics informed loss function for Eq. (7):

$$\mathcal{L}(\theta) = \mathcal{R}(u_\theta) = \left\|\frac{\partial u_\theta}{\partial t} + \frac{\partial u_\theta}{\partial x}\right\|_2^2 + \left\|u_\theta|_{t=0} - \sin(x)\right\|_2^2 + \left\|u_\theta|_{x=0} - u_\theta|_{x=2\pi}\right\|_2^2 \tag{8}$$

where $\|\cdot\|_2$ is the averaged $l^2$ norm, i.e. root mean squared error (RMSE). The loss is trained on a set of 956 collocation points: 256 points for the initial condition, 100 points for each boundary, and 500 points for the interior of the domain where the differential operator is minimized. The initial condition and boundary points are selected to be evenly spaced grids and don't change during training. The interior points are randomly sampled from the domain at each iteration of training. This differs from the typical approach in the PINN literature, where full-batch training is often used. However, we are interested in testing the performance of our approach when using first-order optimization methods, which are known to perform better on small stochastic batches of training data.

We compare the relative errors for each method in Fig. 1. The relative error is computed on a fine grid rather than on the set of collocation points. The PINECONE reaches the minimum error of the PINN at around iteration 1,200. The PINN takes over 22 times as many iterations to reach the same error. We report the minimum, mean, and maximum relative errors for iterations after the PINECONE achieves the PINN's smallest loss in Table 1. The PINECONE outperforms the PINN on each of these metrics.

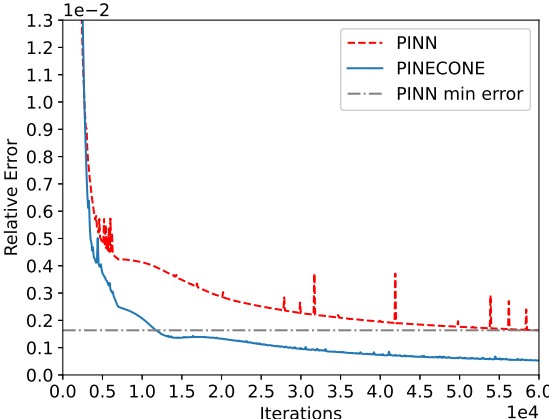

Figure 1: Training iterations v.s. the relative error for a PINECONe (solid blue line) and a PINN (dashed red line) for Eq. (7), evaluated on a fine grid of testing data. The minimum error reached by the PINN approximation is shown as a horizontal line in grey with a dash-dot pattern.

| $\frac{\|u_\theta - u\|_2}{\|u\|_2}$ | Min | Mean | Max |
|---|---|---|---|
| PINN | $1.636e^{-3}$ | $1.947e^{-3}$ | $3.717e^{-3}$ |
| PINECONE | $5.273e^{-4}$ | $7.278e^{-4}$ | $1.067e^{-3}$ |

Table 1: Relative errors for a PINECONE v.s. a PINN for Eq. (7), evaluated on the testing data. Errors are computed only for the iterations after the PINECONE has reached the PINN's minimum training error (see Fig. 1).

## 3.2 BURGER'S EQUATION

We next consider one-dimensional Burger's equation with very small viscosity,

$$\frac{\partial u}{\partial t} + u \frac{\partial u}{\partial x} - \frac{0.01}{\pi} \frac{\partial^2 u}{\partial x^2} = 0, \quad \text{for } (x,t) \in \Omega = [-1,1] \times [0,1],$$
$$u(-1,t) = u(1,t) = 0, \quad \text{for } x \in \partial\Omega = \{-1,-1\},$$
$$u(x,0) = -\sin(\pi x).$$

$$(9)$$

The performance of PINECONEs is compared to that of a standard PINN for this new problem. The networks both have four hidden layers, all with width 40. Both networks were optimized using AMSGRAD (Reddi et al., 2023) instead of ADAM as the former outperformed ADAM in trials comparing both methods. The learning rate is initially set to $1e-02$ and lowered to $1e-4$ after 2,500 iterations, then again to $1e-5$ at 5,000 iterations. The networks are trained for 30,000 iterations. To facilitate training, the input layer of the network is wrapped in a sine function. This simple change was inspired by the idea of Fourier feature embeddings described by (Dong & Ni, 2021; Wang et al., 2022a). Although considerably simpler than a full Fourier embedding, we observed that it nonetheless provided some acceleration to the training for both network architectures. Note that while the loss function is the same, the residual, boundary, and initial conditions shown in Eq. (8), are replaced with the corresponding counterparts from Eq. (9).

In both test problems considered in this paper, PINECONEs demonstrate significantly more rapid initial progress toward learning solutions than PINNs and achieve lower error throughout the optimization process. It is clear from Fig. 2 that by iteration 3,000 the PINECONE has already captured the salient features of the PDE solution for Eq. (9), by iteration 3,000 the PINECONE already resembles the true solution; it has captured the shape and salient features of the PDE. The PINN, on the other hand, has not yet begun to capture the shock. After an initial period of rapid progress, training for the PINECONE slows dramatically. The PINN never achieved error below $1e-01$,

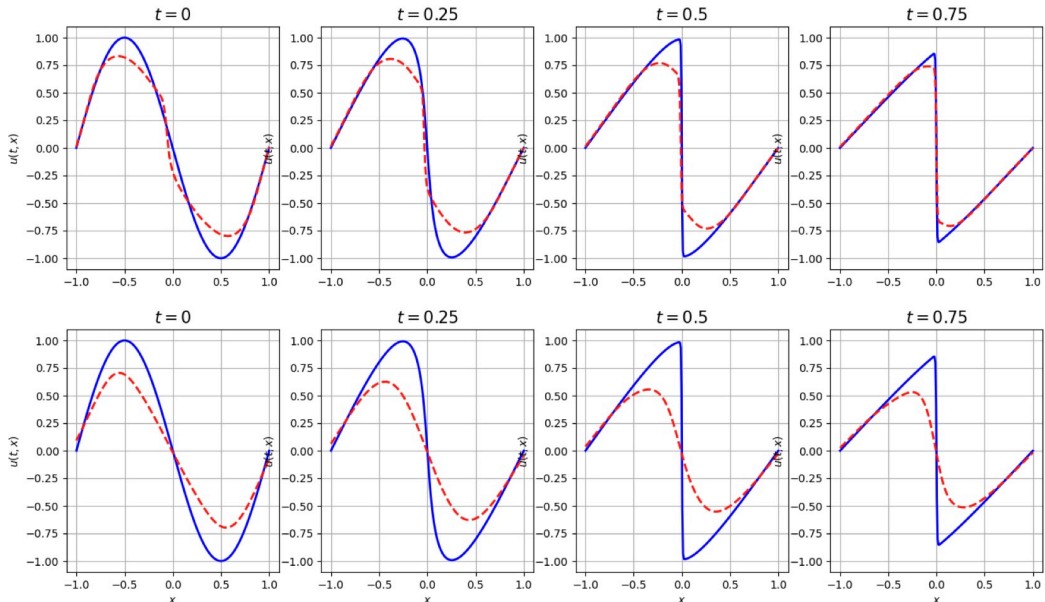

Figure 2: A snap shot that shows differences in training for PINECONEs (top) and PINNs (bottom) at iteration 3,000. Comparison of the approximate v.s. exact solutions are given at four temporal snapshots. The approximations are depicted by the red dashed vertical lines, while the exact solution is shown in solid blue.

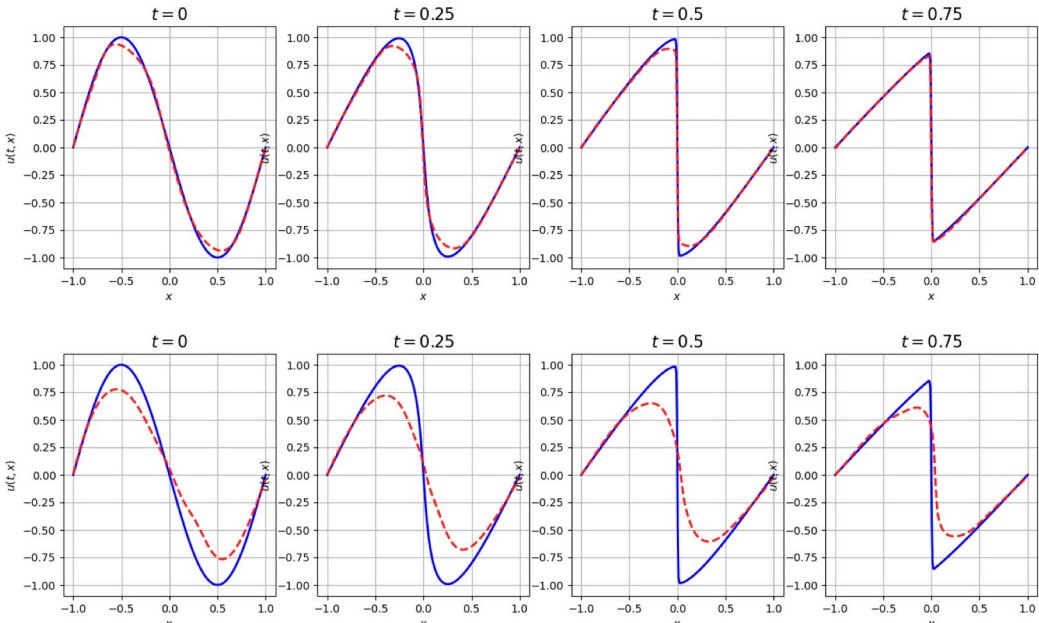

Figure 3: A snap shot that shows differences in training for PINECONEs (top) and PINNs (bottom) at iteration 28,000. Comparison of the approximate v.s. exact solutions are given at four temporal snapshots. The approximations are depicted by the red dashed vertical lines, while the exact solution is shown in solid blue.

while the PINECONE never achieved relative errors below the order of $1e - 02$ in the experiments we performed in this preliminary work. Fig. 3 shows a snapshot near the end of training where

differences between the quality of the approximations continue to be notable. The PINECONE has fully resolved the shock formation but struggles to finish learning the initial condition.

### 3.3 DISCUSSION

Although PINECONEs outperform PINNs in our experiments, training stalls and cannot achieve high accuracy. Studies of PINNs have noted the need to employ L-BFGS to achieve lower error (Raissi et al., 2019a; He et al., 2020; Krishnapriyan et al., 2021), which aligns with observations of stiffness in the gradient flow dynamics for PINNs (Wang et al., 2022b). Future work should compare the training dynamics of PINNs and PINECONEs. While there is a possibility that PINECONEs can alleviate stiffness, it is also possible that the expressive architecture of a PINECONE plays a significant role in its performance. The observed differences in performance may be due to the adaptive depth provided by adaptive time-stepping in a Neural ODE, or to the requirement that the approximate solution satisfy an ODE in $\tau$ for each collocation point rather than indicating lower stiffness.

## 4 CONCLUSION

We develop a formulation that allows the integration of a Neural ODE into a Physics Informed Neural Network task. This expands what PINNs can do by for example, allowing for seamless integration of irregularly sampled time-series data into Physics Informed Learning tasks. We demonstrate that for first-order optimization methods, PINECONEs outperform standard PINNs on two preliminary test problems. Many avenues for improvement remain to be explored by combining active and ongoing research into PINNs with the PINECONE architecture.

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

## A  THE VERIFICATION PROBLEM

In numerical simulations, to test that a numerical solver is free of programming errors that impact the accuracy of the solution, it is common to use the method of manufactured solutions (Roache, 2001). Because analytical solutions are unavailable, the performance of the code is verified on an artificial problem with known solutions. In a similar spirit, we verify that the PINECONE ODE system returns accurate partial derivatives for $\mathbf{z}$ with respect to $\mathbf{z}_0$ by testing against a carefully chosen ODE with known solutions.

We choose the non-linear ODE

$$\frac{d\mathbf{z}}{d\tau} = \begin{cases} \frac{dz^{(1)}}{d\tau} & = -(z_{(1)}z^{(2)})^2 \quad z^{(1)}(0) = x \\ \frac{dz^{(2)}}{d\tau} & = -(z^{(2)})^3 \quad z^{(2)}(0) = t \end{cases}. \tag{10}$$

The dependence of the analytical solution $\mathbf{z} = \left[ \frac{2x}{\ln(2t^2\tau+1)x+2}, \pm\frac{t}{\sqrt{2t^2\tau+1}} \right]^T$ on the initial condition $\mathbf{z}_0 = [x,\ t]^T$, ensures that, at any order, the partial derivatives of $\mathbf{z}$ with respect to $x$ and $t$ are not all zero. Additionally, the system is chosen so that the Jacobian and vector-valued Hessian of $\mathbf{z}$, with respect to $\mathbf{z}_0$, are not diagonal. These two conditions are necessary otherwise, terms in the matrix and tensor products on the right-hand side of Eq. (6) vanish and cannot be checked.

We verify the derivation in Section 2, for a second order PINECONE. Instead of parameterizing $\mathbf{F}_\theta$ with a neural network, for which analytical partial derivatives are infeasible, we set the vector field of $\mathbf{z}$ equal to right hand side of Eq. (10). We then compute $\frac{dS_1}{d\tau}$ and $\frac{dS_2}{d\tau}$ analytically and verify that the equalities in Eq. (6) hold.

The numerical implementation of Eq. (6) is also checked. We randomly generate a set of $(x,t)$ points. The system is solved numerically for each randomly generated initial condition $(x,t)$, using the Dopri8 method from the Diffrax library, a highly accurate Runge-Kutta method. These numerical solutions are compared to their known analytical counterparts, $\mathbf{z}$, $S_1$, and $S_2$ evaluated at the given $(x,t)$. The accuracy of solutions depends on the ODE solver used and the error tolerances chosen. The order of the error for $\mathbf{z}$, $S_1$, and $S_2$ for different choices of ODE solvers with the absolute tolerance fixed at $1e-06$ and the relative tolerance set at either $1e-03$ or $1e-06$ are shown in **??**.

| Errors for verification problem | | | | | | |
|---|---|---|---|---|---|---|
| | **z** | | $S_1$ | | $S_2$ | |
| Tolerance | **1e − 03** | **1e − 06** | **1e − 03** | **1e − 06** | **1e − 03** | **1e − 06** |
| Heun | $1e-06$ | $1e-09$ | $1e-05$ | $1e-08$ | $1e-04$ | $1e-07$ |
| Tsit5 | $1e-07$ | $1e-09$ | $1e-06$ | $1e-08$ | $1e-04$ | $1e-07$ |
| Dopri8 | $1e-12$ | $1e-12$ | $1e-10$ | $1e-10$ | $1e-08$ | $1e-09$ |

## B PINECONE DERIVATION

**First-order PINECONE:** Introducing a new variable $S_1(\tau,\mathbf{z}_0):=\nabla|_{(\tau,\mathbf{z}_0)}\mathbf{z}\in\mathbb{R}^m\times\mathbb{R}^m$ and taking the derivative with respect to $\tau$ of $S_1$ we get:

$$
\begin{aligned}
\frac{\mathrm{d}}{\mathrm{d}\tau}\bigg|_{(\tau,\mathbf{z}_0)} S_1(\tau,\mathbf{z}_0) &= \frac{\mathrm{d}}{\mathrm{d}\tau}\bigg|_{(\tau,\mathbf{z}_0)} \left(\nabla|_{(\tau,\mathbf{z}_0)}\mathbf{z}(\tau,\mathbf{z}_0)\right) \\
&= \nabla|_{(\tau,\mathbf{z}_0)}\left(\frac{\mathrm{d}}{\mathrm{d}\tau}\bigg|_{(\tau,\mathbf{z}_0)}\mathbf{z}(\tau,\mathbf{z}_0)\right) && \text{by Clairout's theorem} \\
&= \nabla|_{(\tau,\mathbf{z}_0)}\mathbf{F}_\theta(\mathbf{z}(\tau,\mathbf{z}_0),\tau)\nabla|_{(\tau,\mathbf{z}_0)}\mathbf{z}(\tau,\mathbf{z}_0) && \text{by the chain rule} \\
&= \nabla|_{(\tau,\mathbf{z}_0)}\mathbf{F}_\theta(\mathbf{z}(\tau,\mathbf{z}_0),\tau)S_1(\tau,\mathbf{z}_0).
\end{aligned} \tag{11}
$$

We can extend the ODE system as many times as desired by iteratively repeating the trick in Eq. (11). To illustrate, let's derive the second-order PINECONE system.

**Second-order PINECONE:** Set $S_2(\tau,\mathbf{z}_0):=\nabla|_{(\tau,\mathbf{z}_0)}S_1\in\mathbb{R}^m\times\mathbb{R}^m\times\mathbb{R}^m$. Taking the derivative with respect to $\tau$ we get:

$$
\begin{aligned}
\frac{\mathrm{d}}{\mathrm{d}\tau}\bigg|_{(\tau,\mathbf{z}_0)} S_2(\tau,\mathbf{z}_0) &= \frac{\mathrm{d}}{\mathrm{d}\tau}\bigg|_{(\tau,\mathbf{z}_0)} \left(\nabla|_{(\tau,\mathbf{z}_0)}S_1(\tau,\mathbf{z}_0)\right) \\
&= \nabla|_{(\tau,\mathbf{z}_0)}\left(\frac{\mathrm{d}}{\mathrm{d}\tau}\bigg|_{(\tau,\mathbf{z}_0)}S_1\right) && \text{by Clairout's Theorem.} \\
&= \nabla|_{(\tau,\mathbf{z}_0)}\left(\frac{\mathrm{d}}{\mathrm{d}\tau}\bigg|_{(\tau,\mathbf{z}_0)}\left(\nabla|_{(\tau,\mathbf{z}_0)}\mathbf{z}\right)\right) \\
&= \nabla|_{(\tau,\mathbf{z}_0)}\left(\nabla|_{(\tau,\mathbf{z}_0)}\left(\frac{\mathrm{d}}{\mathrm{d}\tau}\bigg|_{(\tau,\mathbf{z}_0)}\mathbf{z}\right)\right) \\
&= \nabla|_{(\tau,\mathbf{z}_0)}\left(\nabla|_{(\tau,\mathbf{z}_0)}\mathbf{F}_\theta(\mathbf{z},\tau)\cdot\nabla|_{(\tau,\mathbf{z}_0)}\mathbf{z}\right) \\
&= \nabla|_{(\tau,\mathbf{z}_0)}\left(\nabla|_{(\tau,\mathbf{z}_0)}\mathbf{F}_\theta(\mathbf{z},\tau)S_1\right) \\
&= \nabla|_{(\tau,\mathbf{z}_0)}\left(\nabla|_{(\tau,\mathbf{z}_0)}\mathbf{F}_\theta(\mathbf{z},\tau)\right)\otimes S_1 + \nabla|_{(\tau,\mathbf{z}_0)}\mathbf{F}_\theta(\mathbf{z},\tau)\otimes S_2 && \text{by the product rule} \\
&= \left(\nabla^2|_{(\tau,\mathbf{z}_0)}\mathbf{F}_\theta(\mathbf{z},\tau)\right)\otimes S_1^2 + S_2\otimes\nabla|_{(\tau,\mathbf{z}_0)}\mathbf{F}_\theta
\end{aligned} \tag{12}
$$

Combining Eq. (4), Eq. (11), and Eq. (12) gives a second order PINECONE whose solutions are the Neural ODE together with all first and second-order derivatives of the Neural ODE with respect to the inputs $\mathbf{z}_0$.

**Dimension reduction** The dimension of the Neural ODE solution is mapped to the appropriate PDE dimension using a simple affine transformation of $\mathbf{z}$, $w(\mathbf{z}) := \mathbf{A}\mathbf{z}(\tau,\mathbf{z}_0) + \mathbf{b}$. Finding the Jacobians for $w$ involves some straightforward calculations.

$$
\nabla|_{(\tau,\mathbf{z}_0)}w(\mathbf{z}) = \nabla|_{(\tau,\mathbf{z}_0)}(\mathbf{A}\mathbf{z}+\mathbf{b})\cdot\nabla|_{(\tau,\mathbf{z}_0)}\mathbf{z} = \mathbf{A}^T S_1.
$$

Similarly,
$$
\begin{aligned}
\nabla|_{(\tau,\mathbf{z}_0)}\left(\nabla|_{(\tau,\mathbf{z}_0)}w(\mathbf{z})\right) &= \nabla|_{(\tau,\mathbf{z}_0)}\left(\nabla|_{(\tau,\mathbf{z}_0)}(\mathbf{A}\mathbf{z}+\mathbf{b})\cdot\nabla|_{(\tau,\mathbf{z}_0)}\mathbf{z}\right) \\
&= \left(\nabla|_{(\tau,\mathbf{z}_0)}\left(\nabla|_{(\tau,\mathbf{z}_0)}\mathbf{A}\mathbf{z}+\mathbf{b}\right)\right)\otimes S_1 + \nabla|_{(\tau,\mathbf{z}_0)}(\mathbf{A}\mathbf{z}+\mathbf{b})\otimes S_2 \\
&= \nabla|_{(\tau,\mathbf{z}_0)}(\mathbf{A}^T)\cdot S_1 + \mathbf{A}^T S_2 \\
&= \mathbf{A}^T S_2.
\end{aligned}
$$

