# OpenReview forum: "Physics Informed Neurally Constructed ODE Networks (PINeCONes)"
_ICLR.cc/2024/Conference — Submitted to ICLR 2024_

### Official Review · Reviewer_cWov · 2023-10-20

**Soundness:** 2 fair
**Presentation:** 3 good
**Contribution:** 2 fair
**Rating:** 3
**Confidence:** 4

**Summary:**

This paper proposes Physics Informed Neurally Constructed ODE Networks (PINECONEs), a pipeline to combine the Neural ODE family with physics-informed loss. The authors evaluate this framework on transport equations and Burger’s equations, compared with PINNs. The proposed method shows faster convergence and better accuracy when using first-order optimization methods.

**Strengths:**

- A framework is proposed by combining Neural ODE architectures and physics-informed loss.
- This paper is easy to follow.

**Weaknesses:**

- The idea is not novel. There are already many works investigating the potential of combining neural differential equations with physics-informed loss [1,2,3].

- The baselines are not sufficient. The proposed method is only compared with standard PINNs. There are many variants of the PINN family, which show better performance [4,5,6]. To convince the readers, I think more baselines are expected.

- The proposed method is only tested on 1D problems. There are many successful implementations of PINNs in 2D and 3D cases [4,5,6], but this paper only investigates 1D systems.

---

**Refs:**

[1] Ji, Weiqi, et al. "Stiff-pinn: Physics-informed neural network for stiff chemical kinetics." The Journal of Physical Chemistry A 125.36 (2021): 8098-8106.

[2] Lai, Zhilu, et al. "Structural identification with physics-informed neural ordinary differential equations." Journal of Sound and Vibration 508 (2021): 116196.

[3] O'Leary, Jared, Joel A. Paulson, and Ali Mesbah. "Stochastic physics-informed neural ordinary differential equations." Journal of Computational Physics 468 (2022): 111466.

[4] Cho, Junwoo, et al. "Separable Physics-Informed Neural Networks." arXiv preprint arXiv:2306.15969 (2023).

[5] Wang, Sifan, Hanwen Wang, and Paris Perdikaris. "Learning the solution operator of parametric partial differential equations with physics-informed DeepONets." Science advances 7.40 (2021): eabi8605.

[6] Wang, Sifan, Shyam Sankaran, and Paris Perdikaris. "Respecting causality is all you need for training physics-informed neural networks." arXiv preprint arXiv:2203.07404 (2022).

**Questions:**

Please see my concerns in **Weaknesses**.

---

> ### Author Response · Authors · 2023-11-23
> **Comment 5**
>
> We thank the reviewer for reviewing our manuscript, for their feedback, and for taking the time to provide useful and relevant references.
>
> - We disagree with the reviewer regarding the novelty of our work. None of the three papers referenced use a Neural ODE to solve a continuous PDE system. In [1], only systems of ODEs are considered, and the derivatives employed in the loss function are not computed by extending the ODE system to include forward sensitivities as is done in PINECONEs. In [2], only ODEs are considered, Neural ODEs are not used to approximate the solution to a PDE system. An ODE system with partially known dynamics is augmented with a Neural network component in the vector field. In [3], the authors use Neural ODEs to learn drift and diffusion coefficients for a model, in contrast to our work, which uses a Neural ODE to approximate the continuous solution to a PDE system via a Physics Informed Loss.
>
> - We were unaware of [4] and thank the reviewer for bringing it to our attention. Many variants of PINNs that improve performance are complementary to the PINECONE framework. For example, in [6], the treatment of the loss function is modified to force the model to minimize the PDE residual in a temporally sequential order. This treatment of the loss is entirely compatible with the use of a PINECONE. Given that PINECONEs outperform baseline PINNs without enforcing causality in the loss function, it is plausible that PINECONEs that include this enhancement will do even better.
>
>
> - We agree that it would be valuable to implement PINNs on 2D and 3D problems.
> The primary purpose of this paper is to present a novel architecture for PINNs and to test if the architecture has properties that make it better suited to representing PDEs than a standard Neural Network. As a first comparison, we elected to use well-studied and simple problems that are prevalent in the PINN literature.
>
> [1] Ji, Weiqi, et al. "Stiff-pinn: Physics-informed neural network for stiff chemical kinetics." The Journal of Physical Chemistry A 125.36 (2021): 8098-8106.
>
> [2] Lai, Zhilu, et al. "Structural identification with physics-informed neural ordinary differential equations." Journal of Sound and Vibration 508 (2021): 116196.
>
> [3] O'Leary, Jared, Joel A. Paulson, and Ali Mesbah. "Stochastic physics-informed neural ordinary differential equations." Journal of Computational Physics 468 (2022): 111466.
>
> [4] Cho, Junwoo, et al. "Separable Physics-Informed Neural Networks." arXiv preprint arXiv:2306.15969 (2023).
>
> [5] Wang, Sifan, Hanwen Wang, and Paris Perdikaris. "Learning the solution operator of parametric partial differential equations with physics-informed DeepONets." Science advances 7.40 (2021): eabi8605.
>
> [6] Wang, Sifan, Shyam Sankaran, and Paris Perdikaris. "Respecting causality is all you need for training physics-informed neural networks." arXiv preprint arXiv:2203.07404 (2022).

---

### Official Review · Reviewer_WRLE · 2023-10-29

**Soundness:** 3 good
**Presentation:** 3 good
**Contribution:** 3 good
**Rating:** 6
**Confidence:** 4

**Summary:**

This research describes a network architecture that integrates the neural ordinary differential equation (ODE) and physics-informed constraint loss. They evaluate the framework using the transport equation and Burger's equation, showing fewer training iterations and higher accuracy than original Physic Informed Neural Networks (PINNs).

**Strengths:**

This paper proposes an interesting framework.

**Weaknesses:**

(1) there are no theoretical results.

(2) experimental results are very limited. The baseline PINN is not implemented well. For example, vanilla PINN works fine for Burger’s equation without any issue within a small number of iterations.

(3) experimental results are with the vanilla machine learning training method. Better optimization algorithms for PINN have been developed. For example, you should use [1] to see if the claims still hold with more practical PINN training methods. Because low-dimensional problems can be solved with traditional PDE solvers such as FEM, PINN is not suitable for the cases where the vanilla machine learning training method is sufficient. Therefore, you need to see if the proposed method still makes sense for practical training methods such as [1] that allow PINN to scale well for practical problems.

[1] Tackling the Curse of Dimensionality with Physics-Informed Neural Networks

**Questions:**

(1) Please provide more supportive information for the sentence: “Hybrid modeling frameworks that incorporate neural networks into scientific modeling problems have yielded impressive results.”

(2) Please conduct experiments using SDGD proposed in "Tackling the Curse of Dimensionality with Physics-Informed Neural Networks"

(3) In the experimental result, the authors said both networks have the same number of layers and identical widths. Please provide detailed information on the network configuration.

(4) What are the relationships and benefits of your methods relative to other PINN models? There are several PINN models, such as Augmented Physics-Informed Neural Networks (APINNs) and Extended Physics-Informed Neural Networks (XPINNs). There is no need to compare them in your experiments. But, the authors should mention these versions of PINNs and how the proposed approach fits in the ecosystem of PINNs in the related work section or conclusion.

---

> ### Author Response · Authors · 2023-11-23
> **Comment 4**
>
> We thank the reviewer for their review and for their suggestions and questions which have helped to improve the manuscript.
>
> (1) Using a Neural ODE for the network architecture in a PINN gives rise to some interesting directions for theoretical results. The underlying relationship between the evolution equation for $\tau$ given by the Neural ODE and the PDE being approximated should be explored further but is beyond the scope of this paper.
>
> (2) Please see point 2. in the response titled "Comment 2", as well as bullet point 2 in "Comment 3". Vanilla PINNs rely on higher-order optimization schemes to perform well for Burger's equation.
>
> (3) Even for 1D problems like Burger's equation, PINNs can struggle to train when using first-order optimization schemes. In [2], there is an illustrative example where a PINN struggles to learn a simple boundary value problem using ADAM and for which the use of L-BFGS helps but does not entirely alleviate the training difficulties. Traditional PDE solvers outcompete PINNs in accuracy but have had the benefit of over 100 years of active research and development. PINNs are still a relatively new tool and are likely to improve with time. It is important to address and understand the challenges that PINNs face for more straightforward cases and problems in order to improve and extend the applicability of PINNs. Any optimization scheme that has been developed for PINNs can be applied to PINECONEs and should be complementary to the benefits seen for standard PINNs.
>
> **Questions:**
>
> (1) A citation has been added to support this sentence. Thank you for the suggestion.
>
> (2) We were not aware of this paper. Thank you for bringing it to our attention. The optimization technique presented in [1] is best suited to high-dimensional problems. High dimensional problems are an important application for PINNs but are beyond the scope of this paper which aims to understand the benefits of NODEs for PINN tasks beginning with well-studied benchmark problems.
>
> (3) Details of the network configuration are provided in section 3. In particular, in the second paragraph of section 3.1. Both networks have eight hidden layers, each with a width of 20. The only difference between the networks is the use of a NODE in PINECONEs, i.e. the network parameterizes the vector field for an ODE system, and the final layer size for the PINECONE is different from that of the PINN but mapped to the right dimension by averaging the outputs.
>
> (4) Because a PINECONE is at its core just a PINN with a NODE for the neural network architecture, PINN techniques, like XPINNs that decompose the domain into subdomains, should be complementary to PINECONEs.
>
> [1] Tackling the Curse of Dimensionality with Physics-Informed Neural Networks
>
> [2] https://github.com/PredictiveIntelligenceLab/USNCCM15-Short-Course-Recent-Advances-in-Physics-Informed-Deep-Learning/blob/master/notebooks/OpenChallenges.ipynb

---

### Official Review · Reviewer_Lv3K · 2023-10-30

**Soundness:** 2 fair
**Presentation:** 2 fair
**Contribution:** 2 fair
**Rating:** 3
**Confidence:** 2

**Summary:**

This paper proposes a novel architecture, called PINECONE, that combines the neural ordinary differential equation (neural ODE) with the physics-informed neural network (PINN). The experiments present that the new model improves training performance compared to the standard PINN; the proposed model requires fewer iterations and yields more accurate solutions for the target equations.

**Strengths:**

To the best of my knowledge, this is the first work that tries to combine the neural ODE with the PINN. The new formulation that extends the given neural ODE system with additional differential equations yields an efficient evaluation of derivatives with respect to the input. These derivatives can be used to update the model parameters, i.e., train the model, with the PINN loss. The experiments demonstrate that the proposed model outperforms the standard PINN model. This will be another variant to improve PINN, particularly for dynamics.

**Weaknesses:**

Despite the nice performance improvement presented through the experiments, I find that only two cases are insufficient to validate the model. Please see the original work of both Neural ODE and PINN.

Additionally, I find that the manuscripts need to be further clarified with more elaborate explanations about the proposed formulation and its verification; please see below for more details.

**Questions:**

The paper claims that the calculation of neural ODE’s sensitivity w.r.t. the input is memory-efficient with the proposed formulation. I find that this is an important contribution yet not crystal clear. I guess this may relate to the adjoint sensitivity method proposed by the original neural ODE. A more clarification would be helpful. It would be better to elaborate more on how to solve the extended PINECONE system and how the additional solutions (i.e., the derivatives) are used for training the model.

Sec 3.1 states "The PINECONE reaches the minimum error of the PINN at around iteration 2,700." However, the graph shows that it’s around 1,200. Am I misinterpreting the graph?

For the Burgers’ equation example, the presented performance of PINN is very different from what the original PINN paper shows. I believe that it is because the first-order optimization method was used instead of L-BFGS, which was used in the original one. I’m not sure if this is a fair comparison.

Will the PINECONE architecture be able to handle data-driven discovery tasks as PINN does?

As minor comments, the following typos could be corrected:
- LBFGs
- In Sec 1.2, "... described by a neural network Eq. (3))."
- In Eq. (7), "$\frac{\partial{u}}{\partial{t}} + c\frac{\partial{u}}{\partial{x}}$"
- In Eq. (8), "... $\|| u_\theta|_{t=0} - \sin \||^2_2$ ..."
- In Sec 3.2, "... lowered to 1e-4 after 2,5000 iterations, …"

---

> ### Author Response · Authors · 2023-11-23
> **Comment 3**
>
> Thank you for the review and the helpful suggestions and questions, as well as for taking the time to review and improve the manuscript.
>
> - The memory efficiency in calculating the sensitivities of the system to the inputs are, as you've guessed, inherited from the memory efficiency of the adjoint sensitivity method proposed in the original Neural ODE paper. The extended system presented in PINECONEs returns the forward sensitivities of the NODE to the inputs during the forward pass. For use in a PINN, these are then employed in the portion of the loss function that minimizes the PDE residual and benefit from the memory efficiency of the adjoint sensitivity method during backpropagation.
>
> Thank you for pointing out the typo in section 3.2. The iteration number was incorrect and has been fixed.
>
> - For the Burgers’ equation example, please see point 2. in the response titled "Comment 2". While it is true that it is possible to achieve better performance with a PINN for Burgers’ equation when using L-BFGS, under a first-order optimization, PINECONEs outperform PINNs. In [1], a PINN for Burger's equation is trained using a standard Monte Carlo sampling technique for training together with a specialized sampling strategy developed in the paper, both using first-order optimization. The baseline performance of the PINN is similar to that of the PINN in this paper.
>
> - Because a PINECONE is just a type of PINN with a special Neural ODE architecture, it should be able to handle data-driven discovery in the same way that standard PINNs can. However, it would be good to confirm this in future work.
>
> - Thank you for pointing out the typos; they have been corrected.
>
> [1] Adcock, Ben, Juan M. Cardenas, and Nick Dexter. "CS4ML: A general framework for active learning with arbitrary data based on Christoffel functions." arXiv preprint arXiv:2306.00945 (2023).

---

### Official Review · Reviewer_jF5i · 2023-11-02

**Soundness:** 2 fair
**Presentation:** 3 good
**Contribution:** 2 fair
**Rating:** 3
**Confidence:** 4

**Summary:**

This paper combines Neural ODEs with PINNs to solve PDEs (PINECONE). The average of the outputs of neural ODEs are taken as candidate functions for PINN solutions. PINECONE demonstrates significantly faster speed and lower error compared to the original PINN.

**Strengths:**

1) Originality: PINECONEs provide a continuous solution like PINN, but replace FNN with an ANODE, and store the partial derivatives as system variables.
2) Clarity: The paper is well-written and easy to follow.

**Weaknesses:**

1) The major weakness is the soundness of technical claims and experiments.

* No enough baselines. In Section 1.5, the authors mentioned that some related works apply NODE solvers to PDE. But none of them are compared in experiments.
* No proper dataset. In Section 1.4, it is claimed that PINECONEs are more suitable for real-world data and high-dimensional PDEs. However, all experiments are about low-dimensional synthetic data.

2) The significance of the result is another weakness.
* The number of iterations may be not a practical and fair measure. PINECONEs need less iteration to converge than PINNs, but the CPU time needed for one iteration is apparently different for these two models.
* The constraint of using a first-order optimizer is not necessary for simple PDEs such as Burger's equation. The PINN with L-BFGS is able to achieve high accuracy within a few numbers of iterations and with moderate memory.
* The overall accuracy of PINECONEs may be not satisfactory even in Burger's equation (Fig 2, up-right). It seems that PINECONEs can not learn a shock wave.

3) A minor weakness is some typos
* A missing $\tau$ in the arguments of $F$ in the RHS of Eq(4).
* The large equation in Section 2 paragraph 3 is not numbered, and it is hard to read. The position of the second $=$ is misleading.

**Questions:**

See points 1 and 2 in the Weakness part.

---

> ### Author Response · Authors · 2023-11-23
> **Comment 2**
>
> Thank you for the constructive feedback, and for taking the time to review the manuscript.
>
> 1.
> - Please see the reply at the beginning of "Comment 1".
> - An important motivation for the use of PINNs, is their ability to seamlessly combine real-world data into the physics-informed loss function. For many applications of interest, data sets are large enough that the reliance of PINNs on higher-order schemes may become a bottleneck. This motivated the use of first order schemes in this paper. We find that PINECONEs outperform PINNs under first-order schemes, which is notable for the reasons mentioned above. However, as you have pointed out, our results don't include realistic problems with large data sets, which is an important direction for future work.
>
> 2.
> - The iterations for a PINECONE are more costly than a PINN due to the use of a Neural ODE, making PINECONE training less efficient than PINN training. Although each iteration is more costly, PINECONEs capture salient features of the PDE solution much more quickly in the training process than PINNs (see Fig. 2) and achieve higher overall accuracy under first-order optimization schemes, suggesting that the architecture is better suited to representing the underlying PDE.
>
> - It is known that PINNs benefit from higher-order optimization methods. As you have noted, Burger's equation has been trained to higher accuracy in the original PINN paper using L-BFGS. A compelling explanation of PINN training difficulties is that PINNs can exhibit stiffness in the gradient flow dynamics of the loss. Gradient descent can be seen as a forward Euler discretization of the gradient flow dynamics. From this viewpoint, it makes sense that PINN training can be improved by use of a quasi-Newton method like L-BFGS. A motivation for the choice of using first-order methods in this paper was to test if the PINECONE architecture could improve the performance of PINNs without the need for higher-order optimization schemes. In our results, PINECONEs outperform PINNs under first-order schemes, but do not seem to entirely escape the stiffness issues that plague PINNs. PINECONEs make rapid progress in the early stages of training but then plateau. This suggests that the underlying architecture is better at representing the PDE but that the gradient flow dynamics of the loss are still stiff. The use of higher-order optimization schemes is likely to further improve the performance of PINECONEs.
>
> - Fig. 2 shows an early stage of training for the PINECONE vs the PINN. Comparing the quality of the approximate solutions at this early stage of training shows that there is not yet any suggestion of a shock for the PINN, while the PINECONE has already largely resolved the shock. The figure aims to demonstrate the contrast between the PINN and PINECONE approximations early in the training process. Please see Fig. 3 which has been added to the manuscript and shows later iterations where the PINECONE has fully captured the shock.
>
> 3.
> Thank you for pointing out the typos,  these have been fixed.

---

### Official Review · Reviewer_Btbx · 2023-11-02

**Soundness:** 2 fair
**Presentation:** 2 fair
**Contribution:** 1 poor
**Rating:** 3
**Confidence:** 3

**Summary:**

A physics-informed method is introduced to model the temporal progression of PDEs. It decomposes a given PDE into a system of ODEs to be solved with Neural ODE subsequently. In comparison to PINN, the introduced method learns to simulate simple one-dimensional (transport and Burger's equations) more accurately by using first-order optimizers.

**Strengths:**

_Originality:_ The idea of converting a PDE into a set of ODEs to be solved with Neural ODE seems appealing, but I could not undertand the difference between PINECONE and other methods mentioned under the third bullet point of the related work section.

_Quality:_ The claims are partially supported by experimental results. For example, the superiority of PINECONE over PINN is demonstrated in two experiments. Other claims, such as memory and time efficiency, however, do not find evidence.

_Clarity:_ The manuscript is decently written and organized but would benefit from a clearer framing. For example, it remained unclear to me, whether ANODEs find application here (why are they introduced so explicitly) and what some functions and variables are doing (see questions below).

_Significance:_ The results point into a good direction but need more evidence. In the current state, I do not see how PINECONE finds a wide application and whether it contributes novel insignts to the community.

**Weaknesses:**

1. Unclear whether PINECONE is more efficient in training time and memory consumption. In particular, the application of Neural ODE is quite costly. How does this scale in equations where many ODEs must be solved to find a solution for a PDE?
2. Few experiments on rather simple problems do not seem to be sufficient to demonstrate the superiority of PINECONE over PINN. For example [[1]](https://proceedings.mlr.press/v162/karlbauer22a.html) provides many benchmarks and models, also comparing PINN, which might give a good source for more comparisons.
3. How does PINECONE compare to state-of-the-art methods? As reported in the related work section, there have been proposed numerous (if not hundreds) of modifications to PINN. A demonstration of how these modifications are applied to PINECONE would be of high value to assess whether PINECONE is also superior to more sophisticated PINN variants. Particularly, comparing against Lee & Parish as well as Rackauckas et al. (2021), cited under the third bullet point in related work, would be essential. In the end, it is crucial to assess the quality of PINECONE, how it compares to other methods, and where it actually strugles.
4. How do PINN and PINECONE perform and compare when both optimized with LBFGS? Does PINECONE benefit similarly to PINN from second-order optimization?

**Questions:**

1. In the loss function at the bottom of page 2, what does $s$ stand for, is it the time step and if so, would you mind using $t$ for comprehensibility? Also, what are the arguments to ODESolve?
2. Is the first line in Equation (7) missing an equals 0? That is $\partial u/\partial t + c\partial u/\partial x = 0$?

---

> ### Author Response · Authors · 2023-11-23
> **Comment 1**
>
> We thank the reviewer for their time and attention to our manuscript and their constructive feedback. Comments and questions are addressed below.
>
> **Originality**
> The related works mentioned under bullet 3 both discretize a PDE in order to obtain an ODE system. In PINECONEs, the PDE is not transformed into an ODE system in either of its variables; instead, the continuous solution to the PDE is approximated by a family of Neural ODE Initial Value problems whose initial conditions are points from the PDE's domain.
>
> **Clarity**
> ANODEs have an increased representational range compared to the original Neural ODE formulation and have been shown to require fewer function evaluations during calls to the ODE solver increasing computational efficiency. Experimentally we found that using ANODEs improved the performance of PINECONEs for the two test problems in this manuscript.
>
>
> **Significance:**
> We agree that more experimental evidence would be beneficial and bolster the promising results presented in this initial study. However, combining a physics-informed loss with a Neural ODE is significant because it opens up useful and novel research avenues. The architecture of a Neural ODE is a differential equation in and of itself. There are interesting mathematical questions about if and how this affects the representational properties of the network architecture for the task of approximating a continuous PDE object. Additionally, there are lines of work in the NODE literature that embed structural properties of physical interest into the network architecture (for example, Hamiltonian NODES). This opens up possibilities for embedding conservation properties into the architecture used in a physics-informed optimization problem. The many lines of work that have explored the benefits and properties of NODEs for differential equations and physical systems are of relevance for future applications of PINECONEs.
>
> **Weaknesses:**
> 1. We agree that the manuscript does not adequately address efficiency. Training a Neural ODE is quite costly, and this does create costly overhead for PINECONEs when compared to the original PINN formulation which benefits from the lower overhead of simply employing a standard Fully Connected Feed Forward Neural Network. This is unavoidable when using Neural ODEs. Bringing down the computational costs of Neural ODEs is an active area of research that PINECONEs can benefit from.
>
> The cost of the ODEs scales in the order and physical dimensions of the PDE. For example, a PDE that involves 3rd derivatives will require extending the system of ODEs 3 times; however, computing the entire tensor product is not necessary unless the PDE being approximated involves third-order derivatives in all the PDE variables. Many of the computations required in both the forward and backward pass for the extended ODE system can be parallelized to improve efficiency. Importantly, since the PINN framework does not involve any discretization in time or space, PINECONEs still avoid the scaling issues that arise when high-dimensional problems are discretized and solved using traditional numerical methods.
>
> 2. We agree that more experimental results would be of interest; however, this work focuses on well-studied problems that are found in the PINN literature. We believe that the results presented in this initial study are promising and warrant further investigation. The main purpose of this paper is to present a novel architecture for PINNs and to assess if the architecture is fundamentally better suited to representing PDEs than a standard Neural Network due to the unique properties that being an ODE system imbue it with. We believe that it is valuable, when comparing novel architectures, to begin with well-studied and simple benchmark problems. Exploring many benchmarks and models is an important but larger-scope task.
>
> 3. As the reviewer has noted, there have been numerous proposed modifications to PINNs.  However, achieving accuracy beyond the order of $1e-04$ remains challenging for PINNs. Techniques to improve PINN accuracy and training are complementary to what is presented here and are likely to further improve the performance of PINECONEs similarly to the benefits seen for standard PINNs. State-of-the-art methods are fully compatible with the PINECONE framework.
>
> 4. Please see the response to a similar comment in "Comment 2".
>
>
> **Questions:**
> 1. Yes, $s$ stands for the time step. The variable $s$ is chosen instead of $t$ to make clear the distinction between the time step of the Neural ODE and the variable $t$, which represents time for the PDE system and comes into the ODE system as a set of initial conditions.
> 2. This was a typo that has been fixed.

---

### Meta-Review · Area_Chair_5LFn · 2023-12-06

**Metareview:**

This paper proposes a time-stepping version of physics-informed neural networks where the temporal evolution is modelled with a neural ODE.

All reviewers broadly agree that this paper is too limited in its experimental evaluation and the theoretical motivation to be accepted. While the idea is novel, it is not developed enough, or demonstrated to work usably fast and correctly.

The authors have argued against this in long responses to all reviews, and to me. I understand the frustration of the review process; and since openreview does not impose a limit on communication, it is only natural to keep pushing for one's paper. But the independent reviews are quite consistent in their analysis. I recommend that the authors take the feedback into account and re-submit elsewhere.

**Justification For Why Not Higher Score:**

The reviews are very consistent.

**Justification For Why Not Lower Score:**

N/A

---

### Decision · Program_Chairs · 2024-01-16

Reject